# Microstructure Analysis and Quality Evaluation of Jujube Slices Dried by Hot Air Combined with Radio Frequency Heat Treatment at Different Drying Stages

**DOI:** 10.3390/foods11193086

**Published:** 2022-10-05

**Authors:** Xuedong Yao, Yongzhen Zang, Jiangwei Gu, Haiyang Ding, Yubao Niu, Xia Zheng, Rongguang Zhu, Qiang Wang

**Affiliations:** 1College of Mechanical and Electrical Engineering, Shihezi University, Shihezi 832003, China; 2Key Laboratory of Northwest Agricultural Equipment, Ministry of Agriculture and Rural Affairs, Shihezi 832003, China; 3Key Laboratory of Modern Agricultural Machinery Corps, Shihezi 832003, China

**Keywords:** jujube slices, drying, radio frequency, food processing, microstructure

## Abstract

Jujubes have been favored by consumers because of their rich nutrition and wide use. Hot air drying has been commercially and typically used to prolong shelf life and acquire the dried produce. Jujube slices were dried with hot air combined with radio frequency (RF) at different drying stages, namely, early (0–2 h, E-HA + RF), middle (2–4 h, M-HA + RF), later (4–6 h, L-HA + RF), and whole (0–6 h, W-HA + RF) stages. This study aimed to investigate the effects of different RF application stages on the microstructure, moisture absorption rate, color, and ascorbic acid of jujube slices. Compared with the hot air drying (HA) group, the E-HA + RF group obtained the best results among the experimental groups because it reduced the cells with a roundness of less than 0.4 by 5%. Moreover, the M-HA + RF group showed better results than those of other groups, with an 18.6% and 48.8% reduction in cells for a cross-sectional area less than 200 µm^2^ and a perimeter less than 25 µm, respectively. The minimum total color difference (ΔE = 9.21 ± 0.31) and maximum retention of ascorbic acid (285.06 mg/100 g) were also observed in this group. Therefore, the method of hot air drying assisted by phased RF is viable in the drying industry to improve the quality of dried agricultural products and reduce energy consumption.

## 1. Introduction

Jujubes (*Ziziphus jujuba* Mill.) are fruits native to China and considered an associate of the Rhamnaceae family. Xinjiang, the largest jujube production base worldwide, accounts for more than 50% of the domestic jujube production [1]. The demand for jujubes has steadily increased because of its rich content in nutritional components, such as proteins, dietary fiber, ascorbic acid, and jujube polysaccharide [2]. Although jujube samples can be consumed fresh, dried jujube slices are a by-product of the fruit and favored by consumers due to their long shelf life and unique flavor. Dried jujube samples can be consumed as a dried fruit, compote, and an auxiliary ingredient in soups.

Although the low cost and simplicity of equipment have made hot air drying a commonly used dehydration method for jujube slices, oxidation browning reaction loss of heat-sensitive nutrients and the hardening and crusting of materials in the later drying period must be considered in the drying process [3,4]. The combination of hot air drying and other techniques has been explored to satisfy consumer requirements. Radio frequency (RF) combined with hot air drying is typical. On the one hand, RF heating technology can improve the water transfer rate and shorten drying time due to its characteristics of large penetration depth, rapid heating, and volumetric heating [5]. On the other hand, flowing hot air can rapidly remove water vapor and diffuse it to the surface of jujube slices in the drying process; thus, local overheating or ignition during RF drying caused by high humidity on the stuff surface must be avoided. Existing studies have demonstrated that the application of RF technology can improve the product quality of hot air drying [6,7,8]. For example, the investigation of drying characteristics of inshell hazelnuts dried by combined radio frequency (RF) and hot air (HA), the results showed that when RF heating was applied during or before HA drying, energy efficiency, drying efficacy, and nut quality were enhanced, the anisidine content is higher than the HA ones [9]. As for the jujube, a recent study showed that compared with HA drying, HARF drying showed a shorter drying time, higher drying efficiency and better product quality, that is, higher vitamin C content, smaller color difference and higher total flavonoid content [10].

The materials shrink not only macroscopically with the decrease in moisture content but also change considerably in their internal structure because drying is a complex process of heat and mass transfer. Therefore, focusing only on the sensory value, reabsorbing capability, and shrinkage is insufficient and the microstructure change must be considered because of its importance in the revelation of various drying phenomena. Some studies compared the microstructure of dried samples acquired via different drying methods and analyzed the influencing factors to acquire a proper drying condition [11,12,13,14,15,16]. Moreover, some have highlighted the microstructure variation caused by the change in drying conditions to reveal the quality alternation of dried samples [17,18,19,20,21,22,23,24]. The comparison of microstructures demonstrated that some nontraditional drying technologies can prevent the loss of quality [25,26,27,28,29,30].

These studies qualitatively explored the microstructure of dried agricultural products, but their universality remains unverified and the objectivity of the conclusions is limited. Hence, quantitative research on the microstructure of dried jujube slices requires further investigation.

In order to solve the bottleneck problem of hot air drying, that is, the hardening and crusting of materials in the later drying period, shorten the drying time and improve the dried jujube slices quality, further optimize the RF–hot air combination drying technique, the Hetian winter jujube, which is an important jujube variety grown in Xinjiang, China, was dried using hot air combined with RF at different drying stages (early, middle, later, and whole periods) in this study. The influence of the RF application stage on color parameters, ascorbic acid, and microstructure changing including the cell cross section area, cross section perimeter and roundness were quantitatively investigated. The information generated from this study contributes to developing efficient combined RF–hot air drying technology.

## 2. Materials and Methods

### 2.1. Sample Preparation

Fresh winter jujubes procured from a local market in Shihezi, Xinjiang were inspected carefully to discard spoiled fruit and prevent possible contamination of bacteria or fungi. Samples of the same size (longitudinal diameter of 30 ± 2 mm, transverse diameter of 26 ± 2 mm) were selected to ensure uniformity of physical characteristics of experimental materials. The fruits were then stored in a freezer at 4 °C for 24 h for future use. The initial moisture content of jujube samples was determined according to the oven method at 79.6% ± 1% on the wet weight basis (w.b.) [31].

The following pretreatment was applied before drying. The fresh winter jujubes were taken out of the freezer, washed and wiped when they returned to room temperature, denucleated, and cut into slices (thickness of 10 ± 0.5 mm). Jujube slices were sealed and stored at 4 °C and a humidity of 97% for 24 h and then restored to room temperature (20 °C) for subsequent experimentation.

### 2.2. Drying Method

As shown in Figure 1, the RF drying system (frequency of 27.12 MHz, rated output power of 6 kW) equipped with a hot air drying module for the drying experiment (COMBI6-S, Stray field International Limited, Wokingham, UK), freezer (BCD-267G, Hisense Rongsheng Refrigerator Co., Ltd., Foshan, China), electronic balance (BSM220.4, Shanghai Zhuojing Electronic Technology Co., LTD., Shanghai, China), portable photosynthesis measurement system (Li-6400, Li-COR Corporation, Lincoln, NE, USA), and scanning electron microscope (JSM-6490LV, Japan Electronics Co., Ltd., Tokyo, Japan) were employed in this study.

According to the pre-experiment, convection drying temperature and air speed were 60 °C and 0.5 m/s, respectively. The drying process was divided into early (E-HA + RF), middle (M-HA + RF), and later (L-HA + RF) stages according to the drying rate. RF was applied to each stage and the whole stage (W-HA + RF) separately to investigate the effect of the RF-applying stage on the drying quality and microstructure of jujube slices dried via hot air. The gap between electrodes was 110 mm. The prepared materials were spread evenly on the polypropylene tray with suitable spacing between one another to facilitate vapor diffusion and avoid overheating caused by the contact of edges of jujube slices. The material tray placed in the central position of the bottom electrode is shown in Figure 2. Jujube slices were taken out and weighed every 30 min. Drying was ended when the moisture content of materials dropped below 7%.

### 2.3. Drying Rate Calculation

The drying rate of samples was calculated as follows:*V* = (*M_t_* − *M*_*t*+Δ*t*_)/Δ*t*(1)
where *V* is the drying rate (g/min), *M_t_* is the sample weight at time *t* (g), and *M_t_*_+Δ*t*_ is the sample weight at time *t* + Δ*t* (g).

### 2.4. Moisture Ratio Calculation

Moisture ratio (*MR*) of jujube slices was calculated as follows:
*MR* = (*M_t_* − *M_e_*)/(*M*_0_ − *M_e_*)(2)
where *M_t_* is the moisture content at any time, *M*_0_ is the initial moisture content, *M_e_* is the equilibrium moisture content, and *t* is the drying time (min) [32]. Me values are relatively small with respect to *M_t_* and M0. Hence, the *M_e_* value is neglected, and the *MR* formula is simplified as follows [33]:
*MR* = *M_t_*/*M*_0_(3)

### 2.5. Microstructure Analyses

A 3 mm × 3 mm × 4 mm cuboid was obtained by cutting the dried jujube slice sample with a blade. Microstructures of the sample cross section were observed with a scanning electron microscope after gold spraying. The working voltage was set to 4.0 kV, and magnification was 200 times.

Image J is appropriate for image processing. The microscopic picture was converted into an 8-bit gray format, and the proportion was set for the calculation. Image threshold was adjusted using the gray difference between the target area and its background, and the image was transformed into a binary one. The module called “Analyze Particles” was utilized in the study. Finally, the measurement results were obtained. The interface is presented in Figure 3.

### 2.6. Moisture Absorption Rate Analysis 

A portable photosynthesis measurement system was assembled, and a transparent hose was used to connect the instrument and the sealed beaker, which was filled with 100 mL of distilled water. Dried jujube slices were cut into 2 cm × 2 cm square samples and placed in the leaf chamber. The measurement time was set to 2 h to acquire the moisture absorption rate of jujube slices. The detection principle is shown in Figure 4.

Moist air enters the leaf chamber from one end and is expelled from the other end of the leaf chamber after being partially absorbed by the dried jujube slice. The hygrometer which is a part of the portable photosynthesis measurement system (Li-6400, Li-COR Corporation, USA) detects the inlet and outlet air humidity, and then calculates the moisture absorption rate of the dried jujube slice.

### 2.7. Ascorbic Acid Analysis

Ascorbic acid content (AAC) of dried jujube slices was analyzed with 2,6-di-chlorophenol-indophenol methodology, as described by the AOAC (1995) method [34]. Samples were not exposed to light during all procedures. The results were expressed as mg/100 g of dry weight basic (d.b.).

### 2.8. Color Measurement

Surface color values of fresh and dried jujube samples were measured with a colorimeter (CHROMA METER CR-410, KONICA MINOLTA Company), where *L**, *a**, and *b** values on the device screen represent lightness (0 = black and 100 = white), greenness (−)/redness (+), and blueness (−)/yellowness (+), respectively. The experiment was performed in triplicate, and its mean was considered the experimental result. The total color change (∆*E*) can be expressed as follows:(4)$$\Delta E=\sqrt{{\left(L^{*}-L_{0}^{*}\right)}^{2}+{\left(a^{*}-a_{0}^{*}\right)}^{2}+{\left(b^{*}-b_{0}^{*}\right)}^{2},}$$
where $L_{0}^{*}$, $a_{0}^{*}$, and $b_{0}^{*}$ represent the color values of fresh samples and *L**, *a**, and *b** represent the color values of dried samples [35].

### 2.9. Statistical Analysis

Statistical analysis was performed using SPSS version 20.0 (IBM Corp., Armonk, NY, USA), Origin 2018 (OriginLab Corp., Northampton, MA, USA), ImageJ 2021 (National Institutes of Health, Bethesda, MD, USA), Photoshop 20.0.8 (ADOBE SYSTEMS INCORPORATED, San Jose, CA, USA) and Microsoft Excel 2010 version 14.0 (Microsoft Corporation, Redmond, WA, USA).

## 3. Results and Discussion

### 3.1. Drying Stage Division

As can been seen from Figure 5, 9.5 h was required for only hot air drying, but that time was shortened by the addition of radio frequency. By the preliminary experiments, it can be seen that when radio frequency was applied for 2 h, drying time was reduced by about 6 h regardless of whether radio frequency is added early, middle or late stage in the hot air drying process, which facilitates further experimental research and analysis. Therefore, the hot air drying process can be divided into four parts in which the RF heat treatment will be added, that is the early stage (0–2 h), the middle stage (0–4 h), the later stage (0–6 h), and the whole stage (0–6 h).

As shown in the figure above, after the application of RF, the drying rate of jujube slices was increased, but their curves were not the same, which we think was related to the different moisture content and microstructure of jujube slices in various drying stages. In the early stage of the hot air drying, the moisture content of jujube slices was highest, the microscopic structure was the most complete, and the application of RF promoted the outward transmission of internal moisture, which was beneficial for maintaining the integrity of the moisture transfer channel of jujube slices, so in the following hot air drying process, the jujube dry rate was still faster than the HA group. The M-RF + HA group showed a similar result, but at this time, the moisture content of jujube slices decreased, the heat absorbed by water vaporization was less, and the internal temperature was higher, which further promoted the degradation of cellulose and pectin, making the cell wall thinner and broken [36], as well as the deformation and expansion of channels and holes in some areas as shown in Figure 6d, which made the moisture loss of jujube slices faster. It was similar to the drying results of Azam [37] on lettuce and the RF–hot air combined drying results of Niu on jujube slices [38]. In the late drying stage, the moisture of jujube slices was essentially concentrated in the central area. The application of RF made the temperature of jujube slices rise sharply and the moisture was removed quickly. Therefore, the drying time of the E-HA + RF group is slightly longer than the M-HA + RF and L-HA + RF ones.

### 3.2. Microstructure Analysis Results 

Figure 6 demonstrates that the cells of fresh jujube slices are intact and regular; however, the microstructure of the dried samples shows various degrees of contraction and cell wall degradation. This phenomenon is probably caused by the role of the cell wall in supporting the tissue structure and controlling the moisture exchange; however, constituents of cell wall gradually degraded with time and resulted in impaired cell wall function and contraction of moisture exchange channels. These manifestations reflect the changes in the microstructure of jujube slices.

The cell wall was seriously degraded, and the microstructure shrinkage and collapse were severe in samples with hot air drying (HA). By contrast, the microstructure of the four groups of jujube slices with RF treatment was loose and regular and the cell wall was maintained properly, probably because RF can heat the material as a whole. The rapid increase in temperature will accelerate the evaporation of water and lead to the expansion of cells and effective reduction in the microstructure damage attributed to hot air drying. This finding is similar to the conclusion of Yang on microstructural changes during fruit and vegetable drying [39].

The application of RF in the early stage (E-HA + RF) of jujube slices led to the sudden rise in internal temperature and subsequent overheating. A portion of the heat was responsible for the diffusion and evaporation of moisture, and the remaining portion increased the activity of enzymes for example cellulase [40] and accelerated the degradation of the cell wall that resulted in the shrinkage and collapse of the tissue structure.

The cell of jujube slices in the M-HA + RF group was regular, and the degree of cell wall shrinkage was small, probably because the moisture content of jujube slices was neither high nor low when the RF heat treatment was applied in the middle stage. The majority of heat was used to maintain the internal temperature after absorbing electromagnetic waves and converting them into heat energy. These processes decelerated the degradation of the cell wall and promoted the outward transmission of internal moisture, shortened the drying time, alleviated the shrinkage of the material surface caused by hot air drying, and improved the moisture retention in the exchange channel. These findings are consistent with those of Dai [41] and Meng [42].

The difference in microstructure of L-HA + RF and HA groups was insignificant, probably because the microstructure of the jujube slice was destroyed and the moisture content was low in the late drying stage. The application of RF at this time resulted in less radiofrequency energy absorption and slower internal moisture evaporation rate than those of E-HA + RF and M-HA + RF groups. Hence, applying RF at the late drying stage was detrimental to the maintenance of the microstructure.

The microstructure of jujube slices in the W-HA + RF group was similar to that of the HA group, probably because the application of radio frequency throughout the drying process provided significantly more energy than the requirement for internal moisture evaporation and the excess energy degraded the cell wall composition that resulted in the shrinkage and collapse of the microstructure.

Structural data of the six groups of jujube slice cells were quantitatively analyzed. The results are shown in Figure 7. Hot air drying will produce severe contraction on the cells, and all cell areas are less than 400 µm^2^. The application of RF alleviated the shrinkage. The addition of RF in the early stage can reduce the cells with an area of less than 400 µm^2^ by 18.1%, and large-sized cells with an area greater than 2600 µm^2^ appear because of the rupture and fusion of cells from the drying process (Figure 7a).

The perimeter change is mainly caused by the contraction and deformation of cells during drying. Figure 7b shows that the application of RF, especially in the early and middle stages, can effectively reduce cell atrophy and deformation caused by hot air drying to ensure that cells with a perimeter less than 50 µm reduce from 0.99 to 0.508 and 0.502; notably, even the application of RF in the late and whole stages can reduce this value to 0.668 and 0.660. Moreover, a large cell perimeter is also caused by the rupture and fusion of cells similar to the area change.

The application of RF during early and middle stages reduced cells with roundness less than 0.4 from 0.27 to 0.18 and 0.23 due to hot air drying. Furthermore, RF application during other stages increased this number (Figure 7c).

### 3.3. Moisture Absorption Rate Analysis Results 

The moisture absorption rate of the five groups of dried samples was determined. The results were illustrated in Figure 8. The trend of the moisture absorption rate of all five groups of samples was consistent, that is, the moisture absorption rate decreased with time, the moisture absorption rate in the four groups with applied RF (19.25–3.39%) was higher than that of the HA group (15.16–2.84%), and M-HA + RF demonstrated the maximum moisture absorption rate (19.25–6.79%). Friedman and Kendall tests were then performed. The results demonstrated that a significant difference exists in the water absorption rate among the five groups of samples (Sig. = 0.00 < 0.05). This finding further confirmed that applying RF, especially in the middle stage of hot air drying, can decelerate the jujube slice contraction and properly maintain its cell integrity.

### 3.4. Color Analysis

Visual color characteristics of dried jujube slices and changes in their *L**, *a**, and *b** were compared with those of fresh samples (Figure 9). Different letters above the bars represent significant differences between the data (*p* ≤ 0.05), while the same letter means there is no significant difference between the data. The application of RF can appropriately maintain the original color of jujube slices (except for the L-HA + RF group), that is, ∆*E* values of other RF–hot air combined dried samples were smaller than those of the single hot air-dried ones (Figure 9a). Multiple comparisons were performed to determine whether the application stage of RF exerted a significant effect on the color change of jujube slices. Figure 9b illustrates that RF applied at the whole stage can better maintain the *L** value than other groups because the short period in which jujube slices were exposed to air shortened the oxidation process [43]. Figure 9c represents no significant difference upon *a** between the E-HA + RF group and the W-HA + RF one, while there are significant differences among other groups. Moreover, the application of RF can mitigate the change in the *a** value, and the M-HA + RF group samples achieved the closest value to the fresh one. As shown in Figure 9d, the difference in *b** values between the M-HA + RF and other groups is significant but insignificant among other comparisons. 

### 3.5. Ascorbic Acid Analysis Results

As shown in Table 1, AAC of hot air drying jujube slices was lower than that of four other groups with added RF. Ascorbic acid is unstable and can easily be affected by high temperature and oxygen [44]. Long exposure to air and then oxidation due to long duration of treatment under hot air condition primarily caused the degradation of ascorbic acid. Moreover, loss was minimized when RF was added because the drying time was shortened. The application of RF at different stages caused different ascorbic acid reservation amounts. The M-HA + RF group achieved the highest ascorbic acid reservation amount at 285.06 mg/100 g, followed by E-HA + RF, W-HA + RF, and L-HA + RF. The ascorbic acid content of the raw material is 587.412 mg/100 g. The high moisture content of jujube slices at the early drying stage enhanced the absorption of RF waves. The internal temperature of jujube slices increased rapidly after converting into heat, thereby accelerating the degradation of ascorbic acid. The long drying time when RF was applied during the later stage was detrimental to ascorbic acid retention. Furthermore, the reduced moisture content when hot air drying with additional RF was applied during the middle stage not only prevented overheating in the experiment samples but also facilitated the shortening of the drying time; this phenomenon is conducive to the retention of ascorbic acid [45,46,47,48]. Notably, significant differences were also observed in the ascorbic acid content among the five groups of samples, and the application stage of RF heat treatment exerted a significant effect on ascorbic acid retention (Table 2).

## 4. Conclusions

This study aimed to determine the effects of the application stage of RF heat treatment on the drying kinetics, color, microstructure, and ascorbic acid content of hot air-dried jujube slices. Results showed that the RF application stage significantly influenced the drying behavior of jujube slices. Drying times were significantly reduced regardless of the RF application stage. Applied RF at the early stage can cause the drying rate to remain nearly constant, while the application of RF in the middle or late stage of hot air drying increased the drying rate more obviously. This may be because, when the water content of the jujube slices was low, adding RF was more conducive to raising the internal temperature of the jujube slices. This reduced the temperature difference between the internal and surface of the jujube slices, and thus promoted moisture transfer. Moreover, Adding RF in the early or middle drying stage could better maintain the original cell structure of jujube slices, that is, the cross-sectional area, cross-sectional perimeter, and roundness of the cells were closer to those of fresh jujube slices. At the same time, RF application during hot air drying can improve the moisture effective absorption rate of jujube slices, with M-HA+RF obtaining the highest values among the sample groups. Notably, the minimum total color difference and the maximum retention of ascorbic acid were also observed in this group. The results of this study can provide a theoretical basis for optimizing RF–hot air combined drying processes not only for jujube slices but also other agricultural products.

## Figures and Tables

**Figure 1 foods-11-03086-f001:**
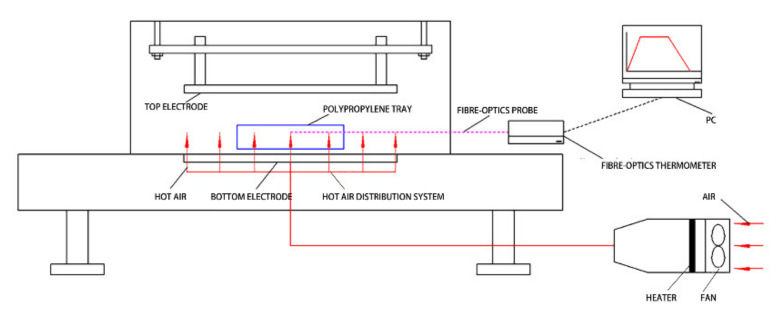
RF–hot air combined drying system.

**Figure 2 foods-11-03086-f002:**
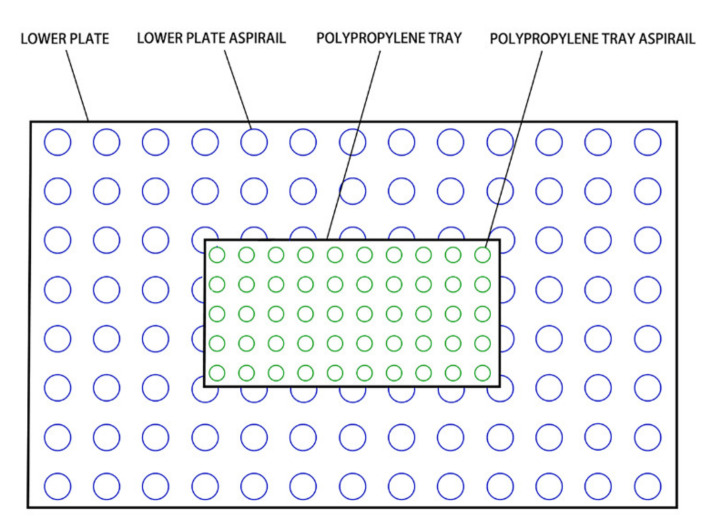
Position of the material tray.

**Figure 3 foods-11-03086-f003:**
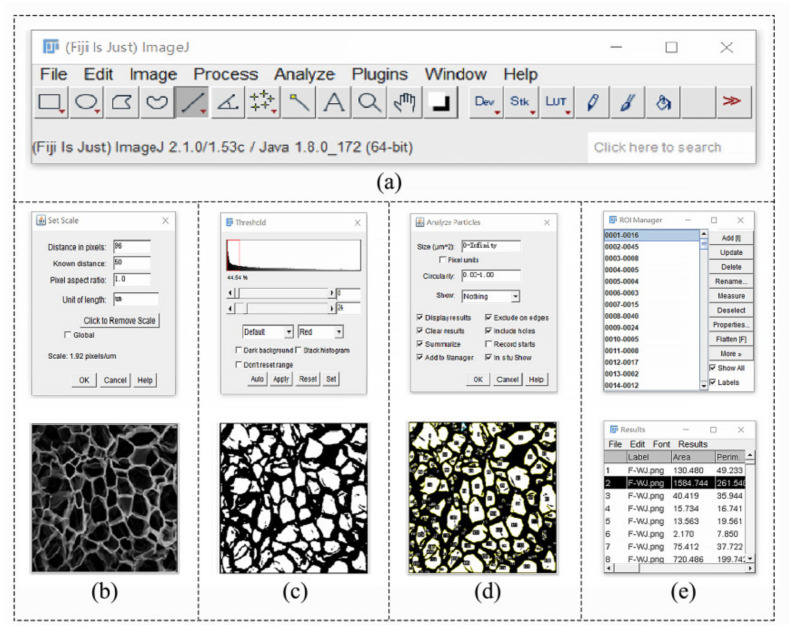
Image process of dried jujube slices ((**a**): main menu; (**b**): scale setting; (**c**): threshold setting; (**d**): particle analysis; (**e**): results).

**Figure 4 foods-11-03086-f004:**
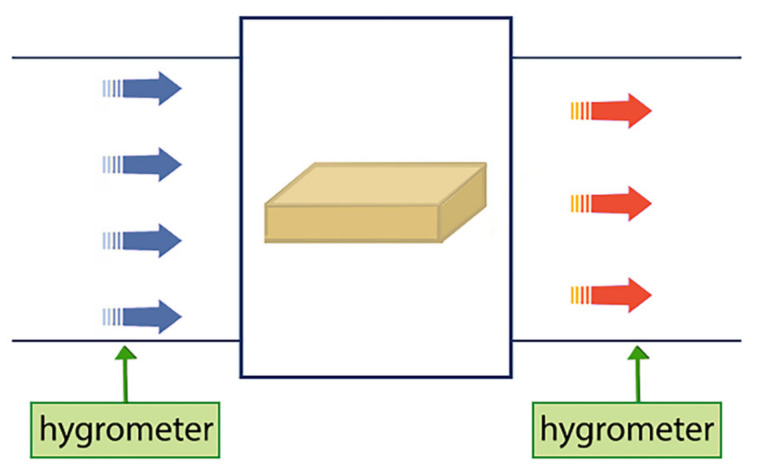
Moisture absorption rate detection principle of dried jujube slice.

**Figure 5 foods-11-03086-f005:**
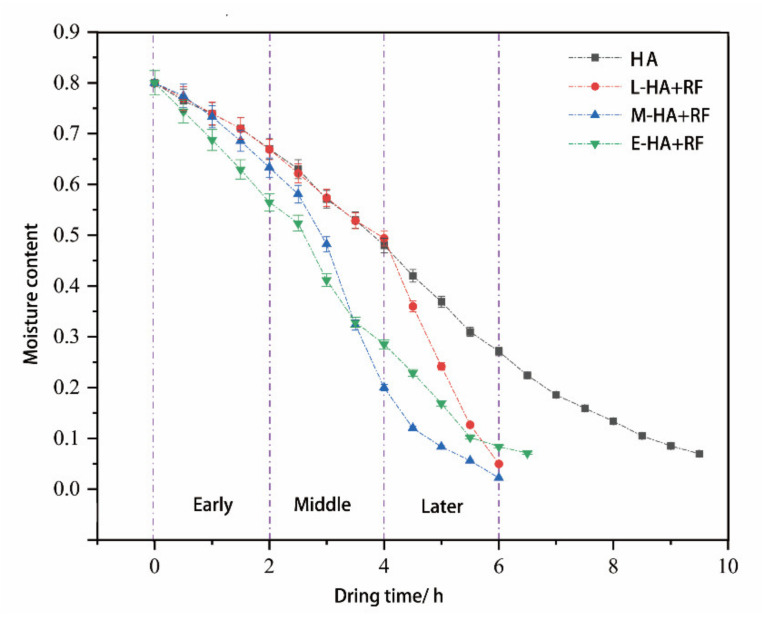
Drying rate of jujube slices at different drying condition.

**Figure 6 foods-11-03086-f006:**
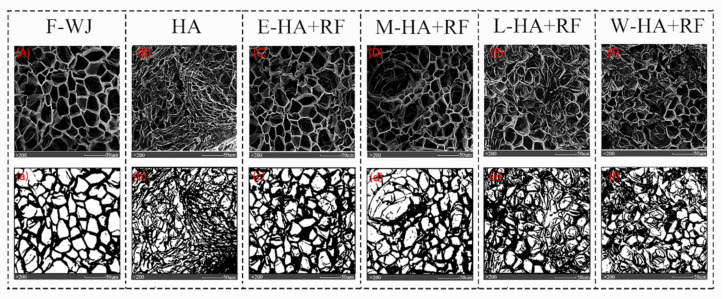
Microstructure and binary images of experiment samples. (**A**) microstructure of fresh jujube slice; (**B**–**F**) microstructure of jujube slices dried by different methods; (**a**) binary image of fresh jujube slice; (**b**–**f**) binary image of jujube slices dried by different methods.

**Figure 7 foods-11-03086-f007:**
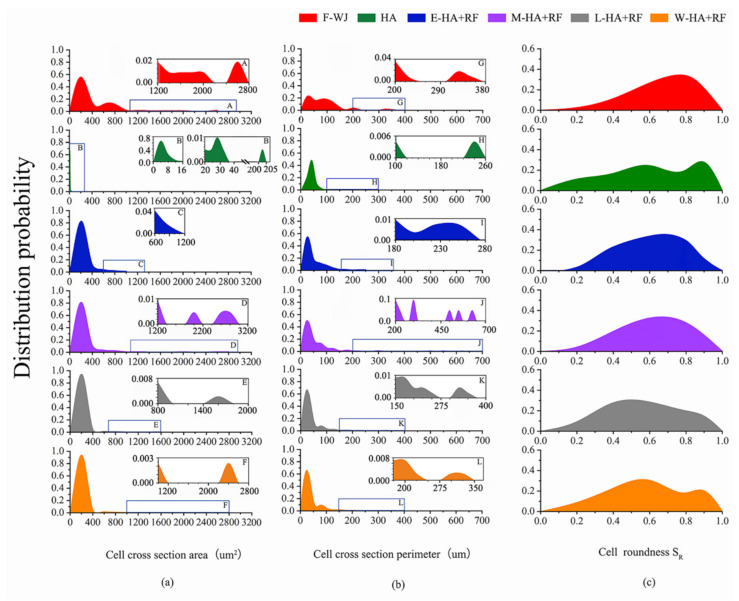
Microstructural data of the six groups of jujube slices ((**a**): cell cross-section area; (**b**): cell cross-section perimeter; (**c**): cell roundness).

**Figure 8 foods-11-03086-f008:**
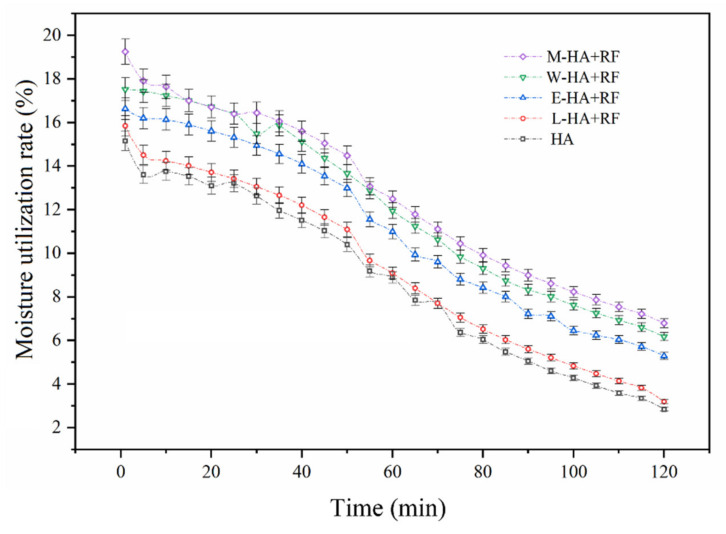
Moisture absorption rate of five groups of dried samples.

**Figure 9 foods-11-03086-f009:**
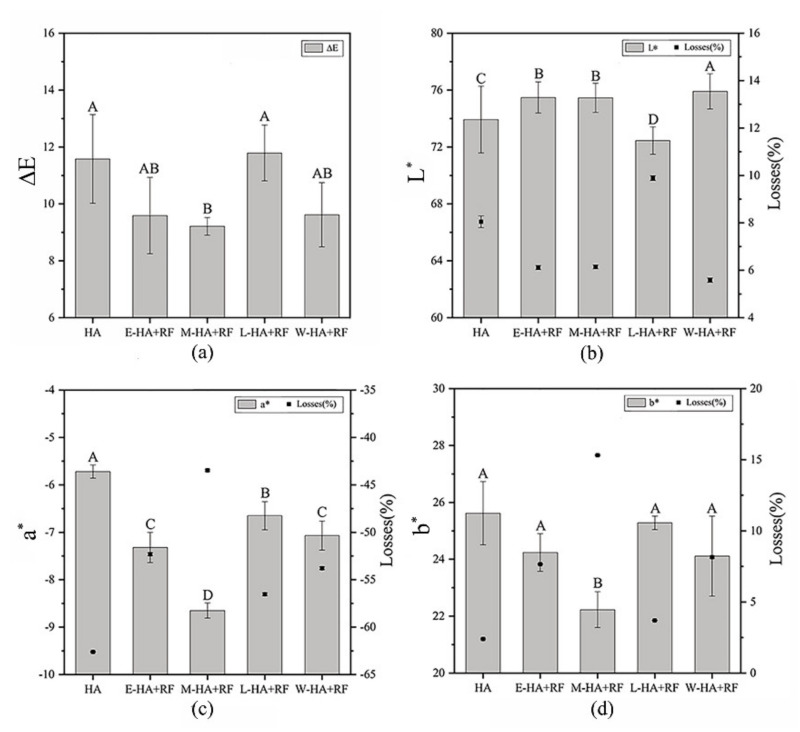
Visual color characteristics of dried jujube slices: (**a**) Δ*E*, (**b**) *L**, (**c**) *a**, and (**d**) *b** values. The same letter above the bars indicates that there is no significant difference between the data, and different letters above the bars indicates that there is significant difference between the data.

**Table 1 foods-11-03086-t001:** Effect of RF heat treatment on AAC of hot air-dried jujube slices.

Groups	Times	Mean Value	Standard Deviation	Standard Error	95% Confidence Interval		Minimum	Maximum
					Lower Limit	Upper Limit		
HA	3	233.53 ^e^	2.95	1.70	226.20	240.85	230.21	235.85
E-HA + RF	3	269.01 ^b^	1.18	0.68	266.08	271.94	268.30	270.37
M-HA + RF	3	285.06 ^a^	1.31	0.75	281.81	288.30	283.55	285.88
L-HA + RF	3	243.90 ^d^	1.16	0.67	241.01	246.78	243.01	245.21
W-HA + RF	3	255.64 ^c^	0.44	0.25	254.56	256.73	255.14	255.90
Total	15	257.43	18.88	4.88	246.97	267.88	230.21	285.88

Significance level: ^α^ = 0.05. It represents that the conclusion error rate for significance testing must be less than 5%. Data with the same letter were considered not significantly different and data without the same letter were considered significantly different.

**Table 2 foods-11-03086-t002:** Multiple comparison of AAC of jujube slices acquired by five drying methods.

Group	N	1	2	3	4	5
HA	3	233.53 ^e^				
L-HA + RF	3		243.90 ^d^			
W-HA + RF	3			255.64 ^c^		
E-HA + RF	3				269.01 ^b^	
M-HA + RF	3					285.06 ^a^
Sig.		1.00	1.00	1.00	1.00	1.00

Significance level: ^α^ = 0.05. It represents that the conclusion error rate for significance testing must be less than 5%. Data with the same letter were considered not significantly different and data without the same letter were considered significantly different.

## Data Availability

The data presented in this study are available on request from the corresponding author.

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
