# Peer review of "Microstructure Analysis and Quality Evaluation of Jujube Slices Dried by Hot Air Combined with Radio Frequency Heat Treatment at Different Drying Stages"

_foods, 2022, doi:10.3390/foods11193086_

Round 1

Reviewer 1 Report

Overall comments: See the specific comments below. The relevant study should be carefully reviewed and discussed. Also, some manufacturing info is missing. See below.

Specific comments:

1.       Line 47-48, a recent study investigated the combination of RF and hot heating to dry the in-shell hazelnuts and improved the nut quality. Since it is relevant to this study, the in-depth discussion should be included.

Development of effective drying strategy with a combination of radio frequency (RF) and convective hot-air drying for inshell hazelnuts and enhancement of nut quality

2.       Line 60, explored

3.       Line 65-67, the objectives of this study should be clearly stated.

4.       Line 80, the samples were dried with an initial temp of 4 C?

5.       Line 88, were employed…

6.       Line 97, gap instead of space, why selected 110 mm?

7.       Line 148, the manufacturing info should be provided for the hygrometer.

8.       Line 171, manufacturing info for each software used in this study.

9.       Line 176, single to only

10.   Line 178, applied

11.   Line 177, could the author show the preliminary data to support the justification for the next step?

12.   Line 194, deleted ‘added’

13.   Line 250, applied RF

14.   The letters should be labeled for the statistical significance of the mean values for different treatment conditions in table 1 and table2.

15.   Line 308, was

16.   Line 315-316, agricultural products with similar structure with jujube?

Reviewer 2 Report

Just minor suggestions:

1) Figure 8, add units on axis X

2) Figure 9. Explain the meaning of capital letters above the bars

3) line 311: what is the meaning of "highest results"? Should be the highest values, to my opinion.

Reviewer 3 Report

The presented manuscript concerns the hybrid drying of jujube slices. The aim of the work is clearly stated; results and discussion confirm its fulfillment. The methodology of the study is presented to a sufficient degree, there is a lack of some minor information, but it does not lower the clarity of the manuscript. 

General comments:

1. Drying jujube slices with RF assistance is quite a new approach. However, there are some papers that were not discussed in this manuscript. For example https://doi.org/10.1590/1809-4430-Eng.Agric.v42n1e20210112/2022 

In my opinion, the authors should review the literature once again and put more emphasis on RF technology. There are more, crucial, parameters that can affect the obtained results for example GAP between the electrodes. Authors should explain the applied parameters better as they reference them in the paper (e.g. temperature profile depends strictly on the gap between electrodes, thus it is particularly important in terms of temperature-dependent parameters - stability of ascorbic acid).

2. RF radiation was applied at different times of the process. In figure 5 we can see that it affects the kinetics of drying differently. But the discussion of the obtained results is very perfunctory. In my opinion, the Authors should deepen this part. The effects are clear.

3. Conclusion section should be duly revised. Currently, this part is just a repetition of the obtained results. Meanwhile, there should be general thoughts about observed phenomena without any numbers. It is difficult to conclude the results and avoid similarities with the abstract but it is essential to work.

Detail comments:

Line 55: Check the formatting of the references.

Line 79: Why sliced jujube was stored for 24 h? 

Line 86-88: Do the SEM and portable photosynthesis system attach to the dryer?

Line 92: Wind? Maybe 'air'?

Line 130: Please provide the required reference for ImageJ software, producer, and version. 

Line 164: Please correct the formula 'b*' instead of 'b'.

Line 171: What statistics were calculated in photoshop? 

Line 203: Is cellulose an enzyme?

Line 258: What is the unit of the X axis?

Line 286-287: Can the authors present the temperature curves? The dryer is equipped with a fiber optic thermometer so I guess that temperature was measured.

Line 296: What was the ascorbic acid content of the raw material? 

Round 2

Reviewer 1 Report

The authors addressed the comments accordingly, except for the objectives should be clearly stated in the introduction part. 

Reviewer 3 Report

In my previous review, I encouraged Authors to deepen the analysis of the influence of RF on drying kinetics. The authors add additional descriptions (lines 184-193) but it is still tough to understand. It is essential as some contradictory statements are present in the literature. Some researchers advise using electromagnetic radiation at the end of drying when the material has a small amount of moisture and the process is controlled by diffusion. Other, stated that microwave or RF should be applied at the beginning of the process, as the material is rich with moisture and radiation may effectively influence the drying. Both strategies have pros and cons and reflect product quality. Fruit and vegetables also behave differently, depending on their structure. In the figure, we see the different courses of drying, depending on the period of application. The final drying time is also different. Why it is so? Ultimately, we evaporated a similar amount of moisture...

The "Drying rate as a function of moisture content" graph could be also beneficial in terms of kinetics assessment. 
